# Generating Equivalent Representations of Code By A Self-Reflection Approach

## Abstract

Equivalent Representations (ERs) of code are textual representations that preserve the same semantics as the code itself, *e.g.,* natural language comments and pseudocode. ERs play a critical role in software development and maintenance. However, how to automatically generate ERs of code remains an open challenge. In this paper, we propose a self-reflection approach to generating ERs of code. It enables two Large Language Models (LLMs) to work mutually and produce an ER through a reflection process. Depending on whether constraints on ERs are applied, our approach generates ERs in both open and constrained settings. We conduct a empirical study to generate ERs in two settings and obtain eight findings. ❶ **Generating ERs in the open setting.** In the open setting, we allow LLMs to represent code without any constraints, analyzing the resulting ERs and uncovering five key findings. These findings shed light on how LLMs comprehend syntactic structures, APIs, and numerical computations in code. ❷ **Generating ERs in the constrained setting.** In the constrained setting, we impose constraints on ERs, such as natural language comments, pseudocode, and flowcharts. This allows our approach to address a range of software engineering tasks. Based on our experiments, we have three findings demonstrating that our approach can effectively generate ERs that adhere to specific constraints, thus supporting various software engineering tasks. ❸ **Future directions.** We also discuss potential future research directions, such as deriving intermediate languages for code generation, exploring LLM-friendly requirement descriptions, and further supporting software engineering tasks. We believe that this paper will spark discussions in research communities and inspire many follow-up studies. The source code and data are available in (Anonymous, 2024).

## 1 Introduction

Given a code snippet, its **Equivalent Representations** (ERs) are textual representations that preserve the same semantics as the code. In this paper, ERs include natural language comments, pseudocode, flowcharts, and more. For instance, natural language comments and pseudocode aid developers in efficiently understanding the code and reducing communication barriers (Geng et al., 2024; Li et al., 2021), while flowcharts visualize the control flow and data flow, facilitating code optimization. These ERs play a crucial role in the software development and maintenance process. Despite their significance, the automatic generation of ERs remains an open challenge.

In this paper, we propose a self-reflection approach to generating ERs of code. Our approach is built upon a dual framework consisting of two Large Language Models (LLMs). Given an input code, two LLMs work mutually and produce an ER through a reflection process. More implementation details of our approach are in Section 2. **Depending on whether adding constraints on ERs, our approach generates ERs in both open and constrained settings.** In the open setting, the form of the ERs is unrestricted, allowing the LLMs to freely use their preferred representations of code. In the constrained setting, we impose specific formats on the ERs (*e.g.,* natural language). *Using this approach, we conduct a large-scale empirical study to generate ERs under two settings, leading to eight key findings.*

**Generating ERs in the open setting.** In this setting, LLMs are free to generate ERs without any constraints. As noted, language is the tool of thought (Mercer, 2000). We believe that the way

LLMs represent code reflects how they comprehend it. Therefore, we analyze the generated ERs and summarize four key findings: ❶ LLMs generate diverse forms of ERs in the open setting, including structured ERs (*e.g.,* dictionaries and tables), natural language ERs (*e.g.,* comments and pseudocode), and symbolic ERs (*e.g.,* arithmetic expressions). ❷ From the structured ERs, we find that LLMs treat the code as a structured sequence instead of plain text. LLMs even can produce a plausible syntax tree. ❸ Based on the natural language ERs, we discover that LLMs can reason about the functionality of APIs based on the code text, generating natural language text to explain the APIs. ❹ Through the symbolic ERs, we find that LLMs use mathematical symbols to represent the numerical calculations in code, converting the code into arithmetic expressions. ❺ For different types of code, LLMs flexibly generate different types of ERs. For example, LLMs often generate SQL-style ERs when the code involves searching for elements in a list. Details of these findings and real cases are in Section 3.1. *These findings offer insights into how LLMs understand various types of code, which in turn helps us explore how to better formulate requirements that align with the way LLMs perceive and generate different types of code.*

**Generating ERs in the constrained setting.** By applying constraints to ERs, our approach can support a wide range of software engineering tasks. In our experiments, we impose specific constraints on ERs, including natural language comments, pseudocode, and flowcharts. Based on the experimental results, we obtain three findings. ❻ Our approach effectively generates ERs tailored to software engineering tasks and significantly reduces hallucinations produced by LLMs. ❼ The generated comments provide clear explanations of the code's purpose and implementation, helping developers quickly grasp the code's functionality. ❽ The generated pseudocode and flowcharts clearly illustrate the control flow and data flow within the code, aiding developers in optimizing the code. Details of these findings and real cases are in Section 3.2. *Looking ahead, researchers can utilize our approach to address various software engineering tasks by applying different constraints.*

Finally, we discuss the future directions of our work.

**Deriving an Intermediate Language for Code Generation.** Directly generating source code from natural language requirements is a challenging task. An effective enhancement strategy is to introduce an intermediate language as a bridge, guiding LLMs to first generate the intermediate language and then the final code (Li et al., 2024; Paul et al., 2024). Our proposed approach can be used to automatically derive an intermediate language. Specifically, we use the approach to obtain an ER that lies between natural language and code, with constraints to ensure it adheres to specific syntax rules. For example, we can generate a table that follows a specific pattern. LLMs can first generate this table to clearly represent the entities and relationships in the requirements, and then proceed to generate the code.

**Exploring LLM-friendly Requirement Descriptions.** Currently, natural language text is primarily used to describe requirements in code generation. However, our findings suggest that in an open setting, LLMs use various forms to represent code. This leads us to consider whether we can express requirements from the perspective of LLMs to facilitate the generation of different types of code. For instance, when a requirement involves extensive numerical calculations, we can directly use mathematical expressions instead of natural language text to describe the requirement. We refer to this approach as "LLM-friendly requirement descriptions." We believe that such descriptions can unlock LLMs' reasoning capabilities and improve their accuracy in code generation.

**Supporting Software Engineering Tasks.** There are many software engineering tasks to generate code-related artifacts, such as code comments and pseudocode. Previous work has primarily focused on specific tasks (Li et al., 2021; Geng et al., 2024). We provide a general and unsupervised approach. On one hand, by adding different constraints, our approach can handle different tasks. On the other hand, it does not require labeled data, making it applicable to tasks where data is scarce.

## 2 THE PROPOSED SELF-REFLECTION APPROACH

This section presents a self-reflection approach to generating ERs of code. We first show an overview of our approach (Section 2.1) and provide detailed descriptions (Section 2.2, 2.3, and 2.4).

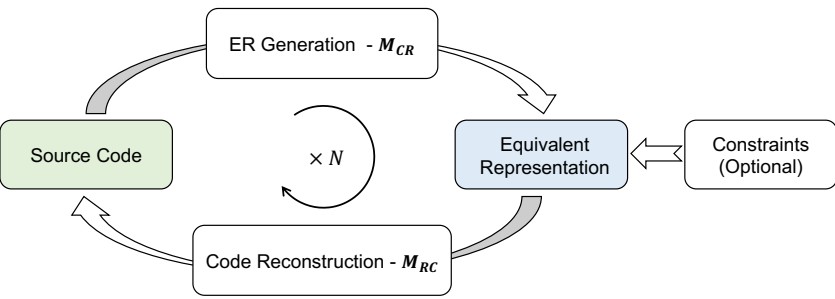

Figure 1: An overview of our self-reflection approach. Two LLMs ($M_{CR}$ and $M_{RC}$) work mutually and produce an ER through a reflection process. We also can add constraints in ERs.

## 2.1 An overview of our approach

Figure 1 shows an overview of our approach. It employs a **dual framework** consisting of two LLMs, *i.e.,* $M_{CR}$ and $M_{RC}$. Given an input code $c$, it produces an ER through multiple trials. In each trial, $M_{CR}$ generates a plausible ER based on the code, and $M_{RC}$ generates a new code snippet $\hat{c}$ based on the ER. The similarity between $c$ and $\hat{c}$ serves as an indicator of how much of the original code's semantics is preserved in the ER. This similarity acts as a supervisory signal to optimize the approach. Besides, our approach allows for external constraints on ERs, which are incorporated into the supervisory signal to guide the process further.

Because LLMs typically are large-scale or black-box, it is heavy and hard to train them further. To address this problem, we follow the Reflexion framework (Shinn et al., 2023) and design a self-reflection algorithm to optimize our approach. Specifically, we convert the numerical supervisory signals (*i.e.,* the similarity between $c$ and $\hat{c}$, external constraints) into **natural language feedback**. The feedback acts as a 'semantic' signal by providing a concrete direction to improve representations. In subsequent trials, the ERs and feedback from previous trials are incorporated as additional context for $M_{CR}$, allowing it to learn from prior mistakes and generate more accurate representations. This mirrors the way humans reflect on past failures to improve their future performance.

## 2.2 Dual Framework

As shown in Figure 1, our approach employs a dual framework consisting of two LLMs, *i.e.,* $M_{CR}$ and $M_{RC}$. Next, we provide a detailed description of two LLMs.

$M_{CR}$ is built upon an LLM capable of following human instructions. Given an input code, we instruct it to transform the code into an ER. The instruction for $M_{CR}$ is shown in Figure 3. If constraints are imposed on the ERs, these constraints are explicitly described within the instruction. As stated in Section 2.1, we expect our approach to learn lessons from past trials. To achieve this goal, we integrate a *memory* module with $M_{CR}$. The memory stores the outputs of the dual framework in past trials. The memory is added as additional context for $M_{CR}$ in the next trial, helping it generate better ERs.

$M_{RC}$ is also an LLM that can follow human instruction. We instruct it to generate a new code snippet based on the ER produced by $M_{CR}$. The instruction for $M_{RC}$ is shown in Figure 4.

## 2.3 Natural Language Feedback

Given an input code $c$, we leverage the dual framework (Section 2.2) to produce an ER and a reconstructed code snippet $\hat{c}$. Since LLMs may generate hallucinations (Zhang et al., 2023), the ER may be wrong. Thus, we evaluate the quality of the ER and provide natural language feedback to guide improvements.

**Scoring the ERs.** An ideal ER should be semantically equivalent to the original code and satisfy external constraints. Thus, we design the following two scoring criteria:

---

**Algorithm 1** The self-reflection algorithm for optimizing our approach.

---

**Inputs:**

   an LLM $M_{CR}$, an LLM $M_E$, Memory $S$, An input code $c$, Threshold $T$, Max_trials

**Outputs:**

   An Equivalent Representation $R$

1: $t \leftarrow 0, S \leftarrow []$
2: $\text{Score}_{sematic} \leftarrow -\infty, \text{Score}_{constraint} \leftarrow -\infty$   *// Initialize evaluation scores*
3: $\text{Best\_Score} \leftarrow -\infty$
4: **while** $\text{Score}_{sematic} < T$ **or** $\text{Score}_{constraint} < T$ **do**
5:    Use $M_{CR}$ to generate a representation $r_t$ based on $c$ and $S$
6:    Use $M_{RC}$ to generate a new code $\hat{c}$ based on $r_t$
7:    Compute $\text{Score}_{sematic}$ and $\text{Score}_{constraint}$ of $r_t$
8:    Produce feedback $f_t$ based on $\text{Score}_{sematic}$ and $\text{Score}_{constraint}$
9:    Push $r_t$ and $f_t$ to $S$
10:    $t \leftarrow t + 1$
11:    **if** $\text{Score}_{sematic} + \text{Score}_{constraint} > \text{Best\_Score}$ **then**
12:       $R \leftarrow r_t$
13:       $\text{Best\_Score} \leftarrow \text{Score}_{sematic} + \text{Score}_{constraint}$
14:    **end if**
15:    **if** $t >= \text{Max\_trials}$ **then**
16:       break
17:    **end if**
18: **end while**
19: **return** $R$

---

❶ *Semantic-equivalent Score.* A semantic-equivalent representation can be seamlessly transformed into the original code. Thus, we measure the similarity between the original code $c$ and reconstructed code $\hat{c}$. This similarity reflects how much of the original code's semantics is preserved in the ER.

How to compute the code similarity remains an open question. Our motivation is that the code not only can be tokenized into a token sequence but also be parsed as a syntax tree. The similarity between code snippets may show in different modalities. As an initial work, we propose a hybrid similarity metric, which combines text similarity $Sim_{text}$ and syntax similarity $Sim_{syntax}$. $Sim_{text}$ measures the rate of overlapping $n$-grams between two code snippets. $Sim_{syntax}$ ignores text surface and computes the rate of overlapping syntax trees between two code snippets. The computations of two similarity metrics are shown in Section A.2.

Finally, we compute the semantic-equivalent score as a weighted sum of $Sim_{text}$ and $Sim_{syntax}$:

$$\text{Score}_{semantic} = a * Sim_{text}(c_o, c_r) + b * Sim_{syntax}(c_o, c_r) \tag{1}$$

where $a$ and $b$ are hyper-parameters. We empirically set them to 0.5 and 0.5.

❷ *Constraint Score.* When additional constraints are imposed on ERs, we further measure whether the ER satisfies the constraints. Since real-world constraints can vary widely, we formally define the constraint score as follows:

$$\text{Score}_{constraint} = F(\text{representation}; \text{constraints}) \tag{2}$$

where $F(\cdot)$ is a function returning a numerical score, which indicates the probability that a representation satisfies the constraints.

**Generating Natural Language Feedback.** We proceed to generate natural language feedback based on the computed scores. As an initial work, we design an empirical feedback template that includes detailed explanations for the scoring criteria and incorporates placeholders for the evaluation scores. This template, illustrated in Figure 5, serves as the foundation for providing comprehensive and coherent feedback. By populating the placeholders with the computed scores, we transform the raw numerical evaluations into well-structured, natural language feedback.

### 2.4 The Self-Reflection Algorithm

Our self-reflection approach can be formalized as an iterative optimization process, outlined in Algorithm 1. Given an input code, our approach enters a loop of trials (Lines 4-18 in Algorithm 1). In each trial, an ER is generated (Line 5), followed by the production of natural language feedback (Lines 7 and 8). These outputs are stored in memory and used as additional context in the subsequent trial (Line 9). The loop terminates when either the ER scores exceed a threshold, or the maximum number of trials is reached. Finally, the ER with the highest evaluation score is returned.

## 3 Empirical Study

Based on whether to add constraints on ERs, we design two settings: an open setting and a constrained setting. We explore generating ERs in two settings and analyze the generated ERs.

**Generating ERs in the open setting (Section 3.1).** We allow our approach to freely generate ERs without constraint. Through a manual inspection of the generated ERs, we have five findings about how LLMs understand the code.

**Generating ERs in the constrained setting (Section 3.2).** We constrain the ERs to be specific forms (*i.e.,* natural language comments, pseudocode, and flowcharts), applying our approach to specific software engineering tasks. Based on the generated ERs, we have three findings showing the effectiveness and flexibility of our approach in supporting various software engineering tasks.

### 3.1 Generating ERs in the open setting

In the open setting, LLMs can freely represent code using their preferred forms. *The language is the tool of thinking.* We think that the forms LLMs represent the code can reflect how they understand the code. We analyze the generated ERs and have five findings. These findings reveal how LLMs understand different elements in code, *e.g.,* grammatical rules, APIs, and numerical calculations.

#### 3.1.1 Study Design

**Datasets.** We select a popular code dataset - CoNaLa (Yin et al., 2018) as the experimental data. It comprises 500 Python code snippets sourced from the well-known programming Q&A platform, Stack Overflow. We consider these 500 code snippets as input data in our experiments.

**Implementation Details.** We select the State-Of-The-Art (SOTA) LLM for code - GPT-4o as the LLMs (*i.e.,* $M_{CR}$ and $M_{RC}$) in our approach. Because our study involves time-consuming human inspection, we first study the SOTA LLM and leave other LLMs in future work. **To prevent our approach from degenerating into a trivial copy model, we impose no constraints on the form of representations except to ensure that they are non-code.** Specifically, we implement the $F(\cdot)$ as a chat with GPT-4o. We instruct GPT-4o to predict a probability score that a representation is non-code. The instruction is shown in Figure 6.

#### 3.1.2 Results and Analyse

We leverage our self-reflection approach to generate 500 ERs in the open setting. We select high-quality ERs whose equivalent score (Equation 5) and constraint score (Equation 2) both exceed 0.9. We manually analyze these ERs and yield the following findings:

> **Finding ❶:** In the open setting, LLMs generate various ERs, including structured ERs (*e.g.,* dictionaries and tables), natural language ERs (*e.g.,* comments and pseudocode), and symbolic ERs (*e.g.,* arithmetic expressions).

**(1) Structured ERs consist of the dictionary, flowcharts, tables, and XML.** Typically, they decompose the code into different components (*e.g.,* APIs, operators, and constants) and store them in structured forms. We show some examples of structured ERs as follows. The percentage in brackets indicates the proportion of the ERs in the dataset.

**(1.1) Dictionary (33.48%), (1.2) Table (7.9%), and (1.3) XML (3.4%).** They represent the code using a JSON-style dictionary, a markdown-style table, and an XML-style text. Each component of the code is mapped to a meaningful key, *e.g.,* `operation_kind` and `Atrtibute`. Three examples are shown in the following listing.

```
Input Code:

result = [int(1000 * random.random()) for i in range(10000)]

ER (Dictionary):

{ "comprehension_kind": "list",
  "element_conversion": "int",
  "element_formula": {
     "operation_kind": "multiplication",
     "operand_1": 1000,
     "operand_2": ...,
  },
  "iteration_var": "i",
  "iteration_range": {"range_kind": "fixed", "range_start": 0, "
     range_end": 10000 }
}

Input Code:

driver.find_element_by_css_selector('someclass').get_attribute('href')

ER (Table):

| DriverContext  | Action | Selector   | Attribute |
|----------------|--------|----------- |-----------|
|      driver    | Find   | 'someclass' | href |

Input Code:

MyModel.objects.update(timestamp=F('timestamp') + timedelta(days
    =36524.25))

ER (XML):

<ObjMethod>
 <ClassName>MyModel</ClassName>
 <MethodName>objects.update</MethodName>
 <Assignment>
  <FieldName>timestamp</FieldName>
  <Expression>
   <FuncCall>
    <FuncName>F</FuncName>
    <Arg>timestamp</Arg>
    <Operator>+</Operator>
    <FuncCall>
     <FuncName>timedelta</FuncName>
     <KwArg> <Key>days</Key> <Value>36524.25</Value> </KwArg>
    </FuncCall>
   </FuncCall>
  </Expression>
 </Assignment>
</ObjMethod>
```

**(1.4) Flowchart (10.26%).** An example is shown in the following listing. Each node in the flowchart denotes a variable (*e.g.*, [Variable "my_float"]) or a operation (*e.g.*, [Replace]). The edge (*e.g.*, <-) between two nodes represents the control flow or data flow. For example, the output of [Cast "float"] is transmitted to [Variable "my_float"].

```
Input Code:

my_float = float(my_string.replace(',', ''))

ER (Flowchart):

[Assign] <- [Variable "my_float"] <- [Cast "float"] ([Replace] <- [
    Variable "my_string"] <- [Arguments [String "," -> String ""]])
```

**(2) Natural language ERs comprise paraphrased APIs, pseudocode, natural language comments, and SQL.** These ERs often use natural language-style phrases or descriptions to represent in code. We show some examples as follows. The percentage in brackets indicates the proportion of the ERs in the dataset.

**(2.1) Paraphrased APIs (30.62%).** An example is shown in the following listing. It paraphrases the APIs in code using natural languages. The paraphrases can reflect the functionality of APIs. For example, it paraphrases the `re.search` into `PATTERN_MATCH`.

Input Code:

```
print(re.search('>.*<', line).group(0))
```

ER (Paraphrased APIs):

```
OUTPUT(TEXT(EXTRACT(PATTERN_MATCH('>.*<', CONTEXT:LINE), PART:0)))
```

**(2.2) Pseudocode (4.5%).** An example is shown in the following listing. It converts the code into a well-formed pseudocode. It typically consists of an explanation of input-output arguments and a multi-step solving process in natural languages.

Input Code:

```
os.kill(os.getpid(), signal.SIGUSR1)
```

ER (Pseudocode):

```
Function: terminate_process
Inputs:
- Library: OS (imported as 'os')
- Library: Signal (imported as 'signal')
Actions:
1. Get current process ID: ('PID')
2. Send signal ('SIGUSR1') to process ('PID')
End
```

**(2.3) SQL (2.7%).** An example is shown in the following listing. It is a SQL-style query statement and uses natural language phrases to denote operations in code, such as `Extract` and `contains`.

Input Code:

```
[i for i, j in enumerate(myList) if 'how' in j.lower() or 'what' in j.
    lower()]
```

ER (SQL):

```
EXTRACT i FROM ENUMERATE myList WHEN j CONVERT_TO_LOWER CONTAINS 'how'
    OR 'what'
```

**(2.4) Natural Language Comment (0.87%).** An example is shown in the following listing. It is a natural language text describing the purpose of code. Interestingly, despite the fact that natural language text makes up a significant portion of LLMs' training data, LLMs rarely use it to describe code. This is an unexpected phenomenon.

Input Code:

```
[(x['x'], x['y']) for x in d]
```

ER (Natural Language Comment):

```
Extract pair (x['x'], x['y']) from all elements in list d
```

**(3) Mathematical ERs mainly contain arithmetic expressions.** They leverage mathematical notations to represent the elements in code, such as APIs and computations.

**(3.1) Arithmetic Expressions (2.2%).** An example is shown in the following listing. It leverages mathematical notation and operators to represent numerical operations in code. For example, `for i in L` $\rightarrow \forall i : i \in L$), and `sum(i)` $\rightarrow \sum(i)$.

---

Input Code:

```
sum(sum(i) if isinstance(i, list) else i for i in L)
```

ER (Arithmetic Expression): $\sum(\sum(i) \leftrightarrow list?(i) \quad \forall i : i \in L)$

---

**(4) Others (3.2%).** The remaining representations do not fit into the above categories and are therefore classified as "Others". These representations typically contain uncommon symbols, as shown in the following example.

---

Input Code:

```
'Sopeton'.encode('latin-1').decode('utf-8')
```

ER (Others): `["Sopeton",` $latin-1 \rightarrow \blacksquare,$ `utf-8` $\rightarrow \blacksquare]$

---

Or, the forms of representations are hard to understand:

---

Input Code:

```
pd.date_range('2016-01-01', freq='WOM-2FRI', periods=13)
```

ER (Others): `<<<DR<<'2016-01-01', 'WOM-2FRI', 13>>>>>`

---

Based on the above ERs, we have some findings about how LLMs understand the code.

> **Finding ❷:** From the structured ERs, we find that LLMs treat the code as a structured sequence instead of plain text. LLMs even can produce a plausible syntax tree.

LLMs generate many structured ERs (55.04%) in the open setting. In the dictionary and table, LLMs parse the code into different components and reason about their types. In the XML, LLMs even produce a plausible syntax tree. These cases show that LLMs view the code as a structured sequence instead of plain text. Thus, we suspect that LLMs can understand the grammatical structures of code. However, existing LLMs do not explicitly model grammatical structures during training. Therefore, how LLMs learn the structures remains unclear.

> **Finding ❸:** Based on the natural language ERs, we discover that LLMs can reason about the functionality of APIs based on code text and generate natural language text describing the APIs.

For example, in the example of pseudocode, LLMs reason about the API's purpose (*e.g.,* `get current process ID`) from its names (*e.g.,* `os.getpid`). This process is akin to the process of reasoning the meaning of a phrase from its abbreviation. Similar phenomena are found in previous studies (Zhang et al., 2022). They found that LLMs can reason about the functionality of code based on a corrupted version.

> **Finding ❹:** Through the mathematical ERs, we find that LLMs use mathematical notations to represent numerical calculations in code and can convert the code into an arithmetic expression.

In the example of arithmetic expressions, LLMs transform code into a mathematical expression. We hypothesize that LLMs leverage mathematical knowledge acquired from other data to aid in understanding code. Mathematical expressions serve to unify various numerical operations within the code, facilitating LLMs' comprehension of underlying data flows.

> **Finding ❺:** For different types of code, LLMs flexibly generate different types of ERs.

LLMs generate different types of ERs based on the features of code. LLMs often generate arithmetic expressions when the code contains many numerical computations. When the code involves

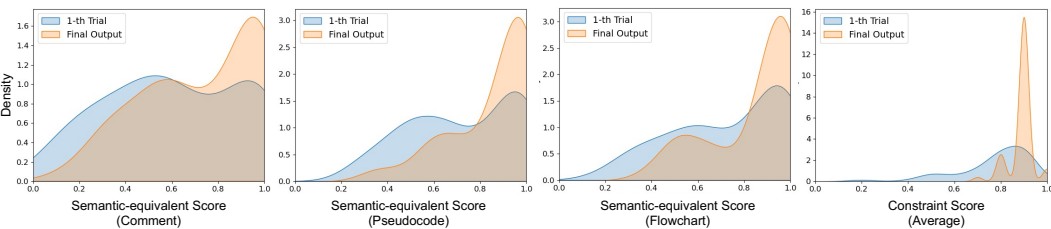

Figure 2: The distributions of semantic-equivalent and constraint scores in three types of ERs. We show the distribution of averaged constraint scores.

searching for elements, LLMs are prone to generate SQL-style ERs. This phenomenon reflects that LLMs can flexibly adopt different ways of thinking based on the characteristics of the code.

## 3.2 GENERATING ERs IN THE CONSTRAINED SETTING

In this setting, we constrain the ERs to be specific forms ( *e.g.,* natural language comments, pseudocode, and flowcharts), applying our approach to specific software engineering tasks (*e.g.,* code comment generation). Through the generated ERs, we obtain three findings, showing the usability of our approach in real-world applications.

### 3.2.1 STUDY DESIGN

We reuse the 500 Python code snippets used in Section 3.1. We select GPT-4o as the LLMs in our approach. Besides, we try three constraints on ERs. First, we constrain the ERs to be a natural language comment. Second, we constrain the ERs to be a well-formed pseudocode. Third, we constrain the ERs to be a clear flowchart. To achieve the above constraints, we implement the $F(\cdot)$ as a chat with GPT-4o. We instruct GPT-4o to predict a probability score that an ER satisfies a specific constraint. The instructions for three constraints are shown in Figure 7, 8, and 9.

### 3.2.2 RESULTS AND ANALYSE

> **Finding ❻:** Our approach effectively generates ERs for software engineering tasks and reduces hallucinations generated by LLMs.

As stated in Section 2.3, our approach scores the ERs in each trial. Figure 2 shows the distributions of final semantic-equivalent and constraint scores in generating three types of ERs. For comparison, we also show the distributions of scores in the first trial. We can see that the ERs in the first trial have lower scores and are often problematic. It may be due to the hallucinations from LLMs, where LLMs generate incorrect ERs though they can understand the code and instructions.

In contrast, the final ERs are substantially improved compared to the ERs in the first trial. We attribute the improvements to our self-reflection algorithm, which enables LLMs to iteratively refine the ERs based on natural language feedback. Figure 10 shows a real case in our experiments, where LLMs continually refine the ERs (*i.e.,* pseudocode) by adding more details.

> **Finding ❼:** The comments generated by our approach explain the purpose and implementation process of code, helping developers quickly understand code.

The following listing shows a natural language comment generated by our approach. The comment accurately describes the code's purpose (`always display warnings`) and the implementation details (`set the warning filter`). It can help developers quickly understand the code and benefit software maintenance.

```
Input Code:    warnings.simplefilter('always')
ER (Natural Language Comment):    Set the warnings filter to always
display warnings
```

**Finding ❽:** The generated pseudocode and flowcharts clearly illustrate the control flow and data flow in code, facilitating developers in optimizing code.

The following listing shows the generated pseudocode and flowchart. They decompose an intricate Python statement into several steps and clearly show the dependencies between these steps. By reading the pseudocode and flowchart, developers can know the control flow and data flow in code, facilitating code optimization (*e.g.,* deleting dead code).

Input Code:

```
sum(sum(i) if isinstance(i, list) else i for i in L)
```

ER (Pseudocode):

```
INITIALIZE total_sum TO 0
FOR EACH i IN L DO
    IF i IS INSTANCE OF list THEN
        total_sum = total_sum + SUM OF ELEMENTS IN i
    ELSE
        total_sum = total_sum + i
RETURN total_sum
```

Input Code:

```
[k for k, v in list(Counter(mylist).items()) if v > 1]
```

ER (Flowchart):

```
       Start
         |
  mylist -> Counter
         |
  Get items of Counter
         |
  Initialize result list
         |
  For each (k, v) in items
    ---------------
    |  Is v > 1?  |
    ---------------
    |     |      |
    No   Yes
          |
      Add k to
    result list - Next item - Output result list - End
```

## 4 RELATED WORK

**Reflexion.** Technically, our self-reflection approach is inspired by the Reflexion framework (Shinn et al., 2023). We follow it and leverage natural language feedback to 'optimize' LLMs instead of updating parameters. In contrast to Reflexion, our approach focuses on generating ERs of code and introduces new effective designs (*e.g.,* semantic-equivalent scores and external constraints). Based on the approach, we further conduct a large-scale study and summarize eight valuable findings.

## 5 CONCLUSION

This paper explores generating Equivalent Representations (ERs) of code by proposing a self-reflection approach. Based on whether to add constraints on ERs, our approach can generate ERs in open and constrained settings. Then, we explore generating ERs in both settings and summarize nine valuable findings. In the open setting, our findings reveal how LLMs understand the different elements in code, *e.g.,* grammatical rules and APIs. In the constrained setting, we find that our approach can effectively support various software engineering tasks. Finally, we discuss the future directions of this work.

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

> Your task is to transform a Python code snippet into a new representation. Only provide the new representation; do not output other explanations. The new representation should satisfy the following principles:
> - Semantic-Equivalent: The representation and the Python code should be semantically equivalent. You can reconstruct the original Python code based on the representation.
> - {Constraint}: {The definition of constraints on representations}
> Here is the Python code:
> {code}

Figure 3: The instruction for representation generation ($M_{CR}$) in our approach.

> You are an experienced Python developer. Please generate a Python statement based on the given representation. Provide only the code; do not output explanations.
> The representation is:
> {representation}

Figure 4: The instruction for code reconstruction ($M_{RC}$) in our approach.

Yue Zhang, Yafu Li, Leyang Cui, Deng Cai, Lemao Liu, Tingchen Fu, Xinting Huang, Enbo Zhao, Yu Zhang, Yulong Chen, Longyue Wang, Anh Tuan Luu, Wei Bi, Freda Shi, and Shuming Shi. Siren's song in the AI ocean: A survey on hallucination in large language models. *CoRR*, abs/2309.01219, 2023. doi: 10.48550/ARXIV.2309.01219. URL `https://doi.org/10.48550/arXiv.2309.01219`.

# A APPENDIX

## A.1 INSTRUCTION TEMPLATES

Figure 3, 4, 5, 6, 7, 8, and 9 show all instructions used in our approach and empirical study. The content in {...} (in blue) is a placeholder. Users can populate these placeholders with specific items during the inference.

## A.2 IMPLEMENTATION DETAILS OF SEMANTIC-EQUIVALENT SCORE

As stated in Section 2.3, we compute the similarity between the original code $c$ and reconstructed code $\hat{c}$ as the semantic-equivalent score. Specifically, we employ a hybrid similarity metric, which combines text similarity $Sim_{text}$ and syntax similarity $Sim_{syntax}$. In this section, we describe the computations of two types of similarities.

**Text Similarity.** Programs with similar functionalities often have similar text surfaces. Researchers have proposed many successful metrics to measure the text similarity between programs Papineni et al. (2002); Eghbali & Pradel (2022). Following existing studies Papineni et al. (2002), we split programs into token sequences and compute the $n$-gram similarity between two code sequences. Specifically,

$$\text{Sim}_{\text{text}}(c_i, c_j) = \exp\left(\sum_{n=1}^{4} \frac{1}{4} \log \frac{s_{ij}^n}{s_j^n}\right) \tag{3}$$

where the maximum of $n$ is empirically set to 4. $s_{ij}^n$ is the number of overlapping $n$-grams between $c_i$ and $c_j$. $s_j^n$ means the number of $n$-grams in $c_j$.

**Syntactic Similarity.** The code is structured and can be parsed into tree structures, *e.g.,* Concrete Syntax Trees (CSTs). Programs with different text surfaces may have similar syntactic structures. Following the related work Ren et al. (2020), we calculate the similarity in syntactic structures. Specifically, we parse the programs into CSTs and extract sub-trees from CSTs. Then, we compute the ratio of overlapping sub-trees between two programs. Formally,

$$\text{Sim}_{\text{syntax}}(c_i, c_j) = \frac{|\text{CST}(c_i) \cap \text{CST}(c_j)|}{|\text{CST}(c_j)|} \tag{4}$$

We have manually evaluated your generated representation and scored it from two aspects:
- Semantic-Equivalent: The representation and the Python code should be semantically equiva-
lent. You can reconstruct the original Python code based on the representation.
- {Constraint}: {The definition of constraints on representations}
Finally, the scores of your representation are: semantic-equivalent score: {semantic_score},
{constraint_score_name}: {constraint_score}.
Based on the above feedback, please further improve the representation to achieve better scores.

Figure 5: The template of natural language feedback.

Please determine whether the given representation is non-code and output a score between 0 and
1. The higher the score, the further the representation is from the source code. Only provide the
score, do not output explanations.
The representation is:
{representation}

Figure 6: The instruction for evaluating whether a representation is non-code.

where $\text{CST}(\cdot)$ is a function for extracting all CST sub-trees of a program.

Finally, we compute a weighted sum of two similarity scores as follows:

$$\text{Score}_{semantic} = a * Sim_{text}(c_o, c_r) + b * Sim_{syntax}(c_o, c_r) \tag{5}$$

where $a$ and $b$ are hyper-parameters. We empirically set them to 0.5 and 0.5.

## A.3 EXAMPLES OF ERs

You are an experienced developer. Please evaluate whether the given representation is a fluent and concise natural language comment, and give it a score between 0 and 1. The higher the score, the better the comment.
The representation is:
{representation}

Figure 7: The instruction for evaluating whether a representation is a natural language comment.

You are an experienced developer. Please evaluate whether the given representation is a well-formed and standardized pseudocode, and give it a score between 0 and 1. The higher the score, the better the pseudocode.
The representation is:
{representation}

Figure 8: The instruction for evaluating whether a representation is a pseudocode.

You are an experienced developer. Please evaluate whether the given representation is a clear and concise flowchart, and give it a score between 0 and 1. The higher the score, the better the flowchart.
The representation is:
{representation}

Figure 9: The instruction for evaluating whether a representation is a flowchart.

Input Code:

```
sum(sum(i) if isinstance(i, list) else i for i in L)
```

ER (1-th trial):

```
SUM(
  FOR EACH element i IN L:
    IF i IS A LIST:
      SUM(i)
    ELSE:
      i
)
```

ER (final output):

```
INITIALIZE total_sum TO 0
FOR EACH i IN L DO
    IF i IS INSTANCE OF list THEN
        total_sum = total_sum + SUM OF ELEMENTS IN i
    ELSE
        total_sum = total_sum + i
END FOR
RETURN total_sum
```

Figure 10: A real case showing the effectiveness of our self-reflection algorithm.