# OpenReview forum: "Generating Equivalent Representations of Code By A Self-Reflection Approach"
_ICLR.cc/2025/Conference — Submitted to ICLR 2025_

### Official Review · Reviewer_vd5R · 2024-10-17

**Soundness:** 1
**Presentation:** 3
**Contribution:** 2
**Rating:** 3
**Confidence:** 5

**Summary:**

The authors propose a dual framework for producing Equivalent Representations (ERs) (e.g., NL comments, pseudocode, flowcharts) for code through multiple trials. In each trial, the first LLM call is used to generate an ER based on the input code and then a second LLM call is used to generate a new code snippet based on the ER. Then, the score is computed between the original code and the new code snippet (as a combination of text similarity, syntactic similarity, and external constraint-based satisfaction score derived from the LLM). This score is then translated into natural language feedback that is used in the subsequent trial to improve the ER. The authors perform this procedure for the 500 Python examples in the CoNaLa dataset and then do a qualitative analysis in two settings: open setting (no constraints on the ER that the model can produce), constrained setting (ER must be of a specific type, i.e., comment, pseudocode, flow chart).

**Strengths:**

- While the dual framework for code isn't itself novel (https://arxiv.org/abs/2310.14053, https://arxiv.org/abs/2402.08699), the idea of translating the similarity metric to NL feedback and performing iterative refinement is quite novel and clever. The iterative approach also seems to help improve equivalence scores, based on Figure 2.
- I particularly liked the idea of scoring based on text similarity, syntactic similarity, and external constraint satisfaction.
- There are many examples provided in the main paper, and the examples in the open setting demonstrate that the LLM is capable of generating many diverse types of ERs based on code.

**Weaknesses:**

- Many of the claims in this paper are not substantiated with empirical evidence. First, in L239-240: "We think that the forms LLMs represent the code can reflect how they understand the code." The authors claim the findings in the open setting "reveal how LLMs understand different elements in code, e.g., grammatical rules, APIs, and numerical calculations." This claim is not supported with any evidence. In fact, just because a model is capable of generating diverse types of ERs doesn't necessarily say anything about whether or not it understands code in that form or that it reasons in that form. For making such a claim, the authors should have run experiments to compare performance when the input or intermediate CoT reasoning chain leverages these different types of ERs. The authors also claim, "From the structured ERs, we find that LLMs treat the code as a structured sequence instead of plain text. LLMs even can produce a plausible syntax tree." However, if the input was something other than code like general text and you still ask it to transform it into some new representation, there's a good chance that it will also be structured. This claim is not well-supported.
- Next, Finding 6 is "Our approach effectively generates ERs for software engineering tasks and reduces hallucinations generated by LLMs." This seems to be based solely on a speculation in L468-470: "We can see that the ERs in the first trial have lower scores and are often problematic. It may be due to the hallucinations from LLMs, where LLMs generate incorrect ERs though they can understand the code instructions." The authors do not provide any evidence of hallucination. In fact, this could also be due to a number of other factors. For instance, it's possible that the LLM didn't capture \textit{all} of the information in the first trial and later added more information. Or that the phrasing or structure of the ER in the first trial was not meaningful to the LLM and it just needed some simple refactoring.
- In Findings 7 and 8, the authors claim that their approach is useful to software developers; however, they do not actually perform any user study to confirm this.
I find that this paper lacks key empirical results. The authors have reported their observations from performing qualitative analysis themselves, but I feel that these are not convincing on their own.
- The authors have only included 1 paper in the Related Work section. I feel that a more thorough literature review is needed. Particularly, there are other papers that use a dual framework for code (https://arxiv.org/abs/2310.14053, https://arxiv.org/abs/2402.08699).

Suggestion:
- The conclusion suggests that future work is presented in detail. However, this is only explained in the introduction. Also, highlighting future work so extensively is not needed in the abstract or introduction. It is better to focus on the approach itself.

**Questions:**

- Is the code and ER not included in the prompt for getting NL feedback (Figure 5)?
- What was the average number of trials? What was the maximum number of trials? How was the threshold score determined?
- Is it possible for the type of ER to change across trials (i.e., in Trial 1 it is a comment, in Trial 2 it is pseudocode)? If so, how often does this happen?

---

> ### Author Response · Authors · 2024-11-26
>
> Thank you for the comments.
>
> ## Response to Weakness 1
>
> Thank you for your suggestion. Exploring the reasoning mechanism inside large language models is an open question. This article provides a novel idea to study this problem and shares some of our findings with the research community.
>
> Our motivation is that language is a tool for thinking [1], and language models perform reasoning by generating tokens, such as chain of thoughts [2]. In the open setting, we allow the large language models to generate a token sequence to represent the semantics of code based on its own preferences. Then, we believe that this generated token sequence (i.e., ERs) can reflect how the models understand this code to a certain extent. Our idea is similar to the Unconditional Positive Regard (UPR) [3] in psychology. UPR creates an environment for patients to express their true thoughts in free speech so that patients can reveal their true thoughts.
>
> In addition, we collected some natural language texts from Wikipedia and used our framework to generate ERs in an open setting. The results show that the generated ERs mainly consist of paraphrased text in other natural languages instead of structured representations. Therefore, we believe that the findings obtained from our experiments are meaningful.
>
> [1] Mercer, Neil. Words and minds: How we use language to think together. Routledge, 2002.
>
> [2] Wei, Jason, et al. "Chain-of-thought prompting elicits reasoning in large language models." Advances in neural information processing systems 35 (2022): 24824-24837.
>
> [3] Bozarth, Jerald D. "Unconditional positive regard." The handbook of person-centred psychotherapy and counselling (2013): 180-192.
>
> ## Response to Weakness 2
>
> Thank you for your comment. During the experiment, we manually checked the correctness of each ER. The dataset we used contained a total of 477 code statements, and our framework generated correct ERs for 399 of them, with an accuracy of 83.65%. For comparison, we manually checked the ERs generated by LLMs in the first trial, of which only 210 code statements had correct ERs, with an accuracy of 44.03%. The main reason for the failure of the initially generated ERs was inconsistency with the original code, including containing wrong information and lacking necessary information. We refer to these causes as LLMs hallucinations.
>
> ## Response to Weakness 3
>
> Thank you for your suggestion. Our work is the first to discuss ERs of code and aims to share some interesting findings we obtained by observing ERs. In addition, we believe that the proposed framework has broad application scenarios. As stated in Section 3.2, developers can use our framework to generate specific ERs, such as natural language comments. As shown in Figure 2, the comments generated by our framework are more accurate than those directly generated by large language models.
>
> In addition, our framework can also be applied to other software engineering tasks. We take code translation as an example to elaborate on the potential of our framework. Code translation aims to convert input code into code in a target language, such as Python-to-Java. Given a code snippet, our framework treats the code in the target language as an ER. Then, we add constraints that the text and grammar of generated ERs must satisfy the target language, for example, they must be successfully parsed by the grammar parser of the target language. Then, our framework can unsupervisedly convert the input code into code in the target language and ensure that the converted code satisfies the lexical and grammatical rules of the target language. As more and more new programming languages emerge, our framework can effectively migrate legacy software to more advanced languages.
>
> ## Response to Weakness 4
>
> Thank you for your suggestion. The two articles you mentioned use a dual framework to evaluate the consistency and code generation capabilities of large models, which is different from the research problem of our work. Therefore, we did not include these two works in the submitted version, and will add them to the revised version.
>
> ## Response to Suggestion 1
>
> Thank you for your suggestion. We will move the Future Work to Conclusion in the revised version.

---

> ### Author Response · Authors · 2024-11-26
>
> Thank you for the questions.
>
> ## Response to Question 1
>
> As we mentioned in Algorithm 1, we set up a memory module to store the original code, ERs generated in previous trials, and natural language feedback. The content in the memory will be fed into the model together with the prompt. Therefore, the prompt in Figure 5 does not contain code and ERs.
>
> ## Response to Question 2
>
> The maximum number of trials is 10, which we determined through preliminary experiments on a small number of samples. About 19% of the samples reached 10 trials. The average number of trials of our framework under different experimental settings in Section 3 is shown in the following table:
>
> | Experimental Setting | Average number of trials |
> |---|---|
> | Open Setting  | 7.49 |
> | Code2Comment | 7.72 |
> | Code2FlowChart | 7.11 |
> | Code2PseudoCode | 8.62 |
>
> ## Response to Question 3
>
> We manually checked the 477 ERs generated in the open setting and found that only 13 ERs satisfy the mentioned condition. The ERs in the initial trials and the ERs in the final trial were of different types. For example, the ER in trial 1 was pseudocode, while the ER in the final trial was an arithmetic expression. We believe such cases reflect that large language models gradually understand the functionality of the code and choose the most suitable ERs to represent it. This phenomenon also validates our motivation that the ERs generated in the open setting can reflect how large language models understand the code.

---

### Official Review · Reviewer_H6fA · 2024-10-22

**Soundness:** 2
**Presentation:** 3
**Contribution:** 2
**Rating:** 5
**Confidence:** 4

**Summary:**

This paper introduces a dual framework to generate the ERs of input code. Experiments demonstrate the effectiveness of the proposed approach and the authors also discuss several potential applicaitions into the software engineering tasks.

**Strengths:**

+ The idea of generating the ERs of code is somewhat interesting.

+ Many vivid examples are provided in the paper.

**Weaknesses:**

- Some doubts about the definitions. What is an ER? The paper merely lists several possibilities without an explixt definition. Also, I wonder can the pseudocode be considered as natural languages, as argued by the authors in the Abstract?

- Unclear about the state of the art in this direction. Is there any approach that aims at generating ERs for a code snippet? What is their limitation?

- Prompt in Figure 4 is not sound. Why statement? I guess it should be a code snippet.

- As for the semantic score, why not directly adopt existing metrics such as CodeBLEU?

- Extended from the above question, I am not convinced that using a metric is reasonable to assess the similarity between c and c'. I guess a code snippet can also have ERs in the form of code. As such, any metric could have brought bias since in this case, c and c' may have different syntactics. In light of this, why not use LLMs to judge code similarities? Why rely on a fixed metric?

- In the dual process, have you considered the inaccuracy brough by the LLMs, especially in the transforming of ERs into code. To what extent is this approach sound?

- Finding 5 is not well investigated. I wonder what features lead to different ERs? Does it mean code functionality or just syntactic features? Any evidence?

- Findings 7 & 8. IMHO, this is an overclaim to conclude them as findings. They are just single cases.

**Questions:**

Please feel free to answer all of my doubts.

---

> ### Author Response · Authors · 2024-11-24
>
> Thank you for the comments.
>
> ## Response to Weakness 1 (What are ERs)
>
> The ER is an abstract concept. Given a code snippet, its ERs can be any textual representations that preserve the same semantics as the code. ERs are often different in different application scenarios. For example, to understand the code, ERs can be natural language comments; to migrate the code, ERs can be functionally equivalent code snippets in other languages.
>
> In our findings, we consider pseudocode as a type of natural language ERs. Our reason is that pseudocode is similar to natural language comments, using natural language to describe operations in the code.
>
> ## Response to Weakness 2 (Unclear SOTA)
>
> We are the first to discuss ERs of code and propose a general framework for generating ERs of code. Thus, there are no state-of-the-art baselines for comparison.
>
> Besides, this paper is not a traditional technical article. We observe the ERs generated by our framework and obtain eight interesting findings. The findings reveal how large language models understand source code. We believe these findings are intriguing and want to share them with the research community. In the future, we will continue to explore the characteristics of LLM-generated ERs, which is of great significance for understanding the reasoning mechanism inside large language models.
>
> ## Response to Weakness 3 (Prompt)
>
> As the early-stage work, we conduct experiments on a statement-level dataset - CoNaLa. Thus, the prompt in Figure is to generate a Python statement. In the future, we will generate ERs for more complex programs, such as functions and classes.
>
> ## Response to Weakness 4&5 (Similarity Metric)
>
> Thank you for the comment. This paper's main contribution is to discuss ERs of code and propose a general framework for generating ERs. Within the framework, how to evaluate the similarity between the original code and the reconstructed code is an open question. As an early-stage work, we conducted experiments on a line-level dataset, CoNaLa. Although the similarity metric we proposed is relatively simple, it can effectively reflect the semantic similarity between two Python statements.
>
> As stated in Section 3.1.2, we select ERs whose semantic-equivalent score (Equation 5) and constraint score (Equation 2) both exceed 0.9 and manually analyze them. We found these selected ERs are correct. The results show that our metric can effectively guide models to generate high-quality ERs.
>
> We also appreciate your suggestion to try more advanced similarity metrics (e.g., CodeBLEU). Our metric can be regarded as a variant of CodeBLEU. In the future, we will try more advanced similarity metrics.
>
> ## Response to Weakness 6 (Inaccuracy brought by LLMs)
>
> Thank you for your suggestion. As the early work, our proposed framework needs further improvement. Currently, we mainly rely on semantic-equivalent scores (Equation 5) and constraint scores (Equation 2) to ensure that ERs are correct.
>
> On the other hand, one of the main contributions is to share eight insightful findings about how large language models understand code. As stated in Section 3.1.2, these findings are concluded from ERs whose semantic-equivalent score and constraint score both exceed 0.9.  We manually observe these ERs and ensure they are correct. Thus, our findings are reliable.
>
> ## Response to Weakness 7 (Different ERs)
>
> By manually observing a large number of ERs, we speculate that code functionality has a greater impact on ERs. For example, code used to find a specific element is more likely to be converted into SQL statements; code containing numerical calculations is often expressed as arithmetic expressions.

---

> > ### Comment · Reviewer_H6fA · 2024-11-26
> >
> > Thanks for the response.
> >
> > I acknowledge that some of my concerns were addressed. Nonetheless, I believe this paper needs significant presentation improvements before it can be published.
> >
> > I have adjusted my score to 5.

---

### Official Review · Reviewer_MRrf · 2024-11-03

**Soundness:** 2
**Presentation:** 3
**Contribution:** 2
**Rating:** 6
**Confidence:** 3

**Summary:**

The authors propose a self-reflection approach utilizing iterative collaboration between two LLMs to generate equivalent representations (ERs) of code snippets. They investigate ER generation under both open and constrained settings. Similarity in text and syntax trees is used to measure program equivalence, leading to the identification and analysis of highly similar ERs, resulting in eight findings. The authors have open-sourced their experimental code and prompts.

**Strengths:**

1. The paper is well-written, allowing readers to clearly understand the methodology, experiments, and corresponding discoveries.

2. The analysis on the CoNaLa dataset provides eight findings that offer insights into how LLMs comprehend code.

3. The availability of open-source code facilitates reproducibility. The code is well-structured, providing clear reproduction steps, data, and prompts.

**Weaknesses:**

1. Novelty: The paper's innovation needs further clarification. The fields of NL2Code and Code2NL have been extensively researched (refer to [1, 2, 3]). What are the core differences between this work and existing studies?

2. Experimental Reliability: Evaluating program equivalence solely through text and syntax tree similarity might be unreliable. The constraint scores are assessed using LLMs; have the authors verified the accuracy and consistency of LLM scoring with human evaluations?

3. The analysis on the CoNaLa dataset yields eight findings, but their significance and guidance for future work are not discussed in the EMPIRICAL STUDY or FUTURE WORK section, suggesting the paper might be incomplete.

```
[1] Hu, Xing, et al. "Deep code comment generation." Proceedings of the 26th conference on program comprehension.2018.
[2] Li, Zheng, et al. "Setransformer: A transformer-based code semantic parser for code comment generation." IEEE Transactions on Reliability 72.1 (2022): 258-273.
[3] Zan, Daoguang, et al. "Large language models meet nl2code: A survey." arXiv preprint arXiv:2212.09420 (2022).
```

**Questions:**

- The paper only analyzes ERs for Python code. Does the analysis of different programming languages ​​show consistency? This may require further experimental evaluation.

- Can the authors elaborate on the impact of their findings on actual software development?

---

> ### Author Response · Authors · 2024-11-24
>
> Thank you for the comments.
>
> 1. This paper explores a general approach for generating equivalent representations of code. Natural language comments are just one type of common equivalent representation. As shown in Section 3.2, our proposed approach can also generate pseudocode and program flow charts for code.
>
> 2. Our paper is not a conventional technical paper. By observing the generated equivalent representations, we gained some interesting findings about how large language models understand source code. We believe these findings are intriguing and want to share them with the research community. In the future, we will discuss the impact of our findings on actual software development.

---

> > ### Comment · Reviewer_MRrf · 2024-11-24
> >
> > Thank you for the clarifications provided in your rebuttal.
> >
> > I appreciate the discussion on the general approach for generating equivalent representations of code and the insight into generating pseudocode and program flow charts. Your work contributes some  findings regarding how large language models understand source code, and I believe this could be a interesting addition to the research community.
> >
> > However, I noticed that the question regarding the experimental reliability—specifically, the accuracy and consistency of the LLM scoring compared to human evaluations—was not fully addressed. This is an important concern, as the reliability of your findings hinges on validating LLM-based assessments against more established evaluation methods.
> >
> > Based on the potential of your work, I have updated my score to 6. However, my confidence adjusted to 3, reflecting the need for further clarification on experimental robustness. I look forward to any additional comments or elaborations on this matter in future revisions.

---

> > > ### Author Response · Authors · 2024-11-24
> > >
> > > Thank you for your feedback and recognition of our work.
> > >
> > > ## Response to Q1 (Experimental reliability)
> > >
> > > It is unclear how to evaluate the similarity between the original code and the reconstructed code. As an early-stage work, we conducted experiments on a line-level dataset, CoNaLa. Although the evaluation metric we proposed is relatively simple, it can effectively reflect the semantic similarity between two Python statements and help generate high-quality ERs.
> > >
> > > In addition, we use a powerful model, GPT-4o, to calculate constraint scores. The constraints imposed in this experiment are relatively simple, such as judging whether an ER is a fluent natural language comment. Our preliminary experiments show that GPT-4o can accurately give constraint scores in our settings.
> > >
> > > Finally, as stated in Section 3.1.2, we select ERs whose semantic-equivalent score (Equation 5) and constraint score (Equation 2) both exceed 0.9 and manually analyze them. We found these selected ERs are correct. The results show that our semantic-similarity scores and constraint scores can effectively guide models to generate high-quality ERs.
> > >
> > > ## Response to Q2 (Future work)
> > >
> > > As mentioned in the last few paragraphs of the Introduction, we believe that ERs of code have at least three uses.
> > >
> > > First, ERs (e.g., pseudocode, natural language annotations, flowcharts) can be used as the chain of thoughts for code generation to improve the accuracy of large language models in code generation. Many works empirically design the chain of thoughts in the code generation process, such as self-planning (Jiang et al. TOSEM) and structured chain-of-thought (Li et al. TOSEM). The ERs generated in this paper are output by the large language model itself, which can better reflect the reasoning mechanism of the model itself. I believe that using such ERs as the chain of thoughts is more conducive to unlocking the reasoning ability of the large language model in code generation.
> > >
> > > Furthermore, by analyzing the ERs, we can detect what ways the large language model prefers to use to understand the code. These findings are helpful for us to design a model-friendly requirement modelling language; for example, we can use tables to write requirements.
> > >
> > > Finally, by adding different constraints, users can convert code into specific forms of ERs, such as natural language comments and pseudocode. Users can also constrain ERs to satisfy specific grammar and use our framework to achieve unsupervised code translation. For example, converting Python code into a programming language code that a large language model has never seen before.

---

### Official Review · Reviewer_3DjZ · 2024-11-04

**Soundness:** 2
**Presentation:** 3
**Contribution:** 1
**Rating:** 3
**Confidence:** 5

**Summary:**

This paper conducts an empirical study to explore the ability of LLMs to generate Equivalent Representations (ER) of code. The authors employ the Reflexion framework to ask LLMs to generate ERs in both open and constrained settings and analyze the generated ERs. Overall, they find that LLMs have the ability to generate ERs for codes, which sheds light on the future exploration of this direction to support code-related tasks.

**Strengths:**

1. The paper is well-written and easy to follow.
2. Discussing ER in cod-related tasks is an interesting topic and remains largely underexplored.

**Weaknesses:**

1. In this paper, the authors stated that LLMs 'can' generate ERs for codes, which are generally cliches. Instead of studying whether LLMs are able to generate ERs, we are more interested in 'how well' LLMs can generate ERs. However, in this paper, there is no quantitative analysis of the correctness of the generated ERs. The semantic-equivalent score and constrained score can not reflect the correctness. The former only focus on semantic similarity, yet a similarity does not guarantee correctness. And the latter is generated by LLMs, which suffers from hallucination problems. The authors should manually check the correctness of generated ERs.
2. The findings lack in-depth analysis. In section 3, basically, there is only a case and a description of it for each finding.
3. The approach employed in this paper is nothing new compared with Relflexion, except for the designed scores for evaluating semantic similarities.
4. The future direction part lacks details and in-depth discussions. After reading this part, I am still confused about how ERs could actually get involved in the three discussed future directions.

**Questions:**

1. For the proposed semantic-similarity score, what is the difference between other famous code similarity evaluation metrics like CodeBLEU and it? Why is it necessary to propose a new metric?

---

> ### Author Response · Authors · 2024-11-24
>
> Thank you for the comments.
>
> ## Response to Q1 (Comparison with CodeBLEU):
>
> Thank you for the comment. This paper's main contribution is to discuss equivalent representations of code and propose a general framework for generating equivalent representations. Within the framework, how to evaluate the similarity between the original code and the reconstructed code is an open question. As an early-stage work, we employ a straightforward approach (i.e., semantic-similarity score), which combines text similarity with syntax similarity. Although the score we proposed is relatively simple, it can effectively reflect the semantic similarity between code snippets and help generate high-quality equivalent representations. Based on these equivalent representations, we obtained many interesting conclusions that reveal how large models understand source code.
>
> We also appreciate your suggestion to try more advanced similarity metrics (e.g., CodeBLEU). Our semantic-similarity score can be regarded as a variant of CodeBLEU. In the future, we will try more advanced similarity metrics.
>
> ## Response to Q2 (Lack quantitative analysis of the correctness of the generated ERs)
>
> We manually checked all generated ERs and only analyzed the correct ERs further. Therefore, I believe that the eight findings in this paper are reliable.
>
> In addition, this paper is not a traditional technical article and does not aim to propose a complex method to improve the ability of large language models to generate ERs. We propose a general framework for generating ERs and obtain eight interesting findings by observing the ERs generated by large language models. The findings reveal how large language models understand source code. We believe these findings are intriguing and want to share them with the research community. In the future, we will continue to explore the characteristics of LLM-generated ERs, which is of great significance for understanding the reasoning mechanism inside large language models.
>
> ## Response to Q3 (Lack in-depth analysis)
>
> Our findings show how large language models understand source code. As stated in finding 1), we found that LLMs mainly generate three types of ERs, including structured ERs (e.g., dictionaries and tables), natural language ERs (e.g., comments and pseudocode), and symbolic ERs (e.g., arithmetic expressions). In finding 2), 3), and 4), we analyze how large language models understand the syntax structures, APIs, and numerical calculations in code. We believe these findings are insightful and help practitioners know the reasoning mechanism inside large language models.
>
> ## Response to Q4 (Comparison with Reflexion)
>
> This paper's key contribution is to discuss the equivalent representation of code for the first time and share eight insightful findings observed from the equivalent representations. As we mentioned in Section 2.1, we were inspired by Reflexion to design this framework. Although our framework is relatively straightforward in terms of technology, it can indeed generate high-quality equivalent representations. We show many ERs generated by our framework in Section 3, based on which we have obtained many interesting findings.
>
> ## Response to Q5 (Future Work)
>
> As mentioned in the last few paragraphs of the Introduction, we believe that equivalent representations of code have at least three uses.
>
> First, equivalent representations of code (e.g., pseudocode, natural language annotations, flowcharts) are used as the chain of thoughts for code generation to improve the accuracy of large language models in code generation. Many works empirically design the chain of thoughts in the code generation process, such as self-planning (Jiang et al. TOSEM) and structured chain-of-thought (Li et al. TOSEM). The equivalent representations generated in this paper are output by the large language model itself, which can better reflect the reasoning mechanism of the model itself. I believe that using such equivalent representations as the chain of thoughts is more conducive to unlocking the reasoning ability of the large language model in code generation.
>
> Furthermore, by analyzing the equivalent representations, we can detect what ways the large language model prefers to use to understand the code. These findings are helpful for us to design a model-friendly requirement modelling language; for example, we can use tables to write requirements.
>
> Finally, by adding different constraints, users can convert code into specific forms of equivalent representations, such as natural language annotations and pseudocode. Users can also constrain equivalent representations to satisfy specific grammar and use our framework to achieve unsupervised code translation. For example, converting Python code into a programming language code that a large language model has never seen before.

---

> > ### Comment · Reviewer_3DjZ · 2024-11-25
> > **Response to Authors**
> >
> > I would like to first thank the authors for their clarification. After carefully reading their responses, I decided to keep my score due to the following reasons:
> >
> > 1. I am still confused about the motivation for proposing a new metric. How does it outperform existing metrics in this topic?
> > 2. I am not criticizing the authors for not improving the performance of generating ERs. I am expecting a quantitative analysis of the accuracy of generating ERs. At least, the authors should provide how many ERs are correct from the 500 generated ERs. An even more detailed quantitative analysis is also needed, like the accuracy of different types of ERs.
> > 3. The authors' responses do not change my opinion on lacking in-depth analysis. For example, the authors only find that LLMs mainly generate three types of ERs, including structured ERs (e.g., dictionaries and tables), natural language ERs (e.g., comments and pseudocode), and symbolic ERs (e.g., arithmetic expressions). A more in-depth analysis may include exploring which types of ERs  LLMs tend to generate for which types of codes.
> > 4. The authors' responses for future directions are still quite general. As an empirical study paper, readers expect more detailed and insightful discussions about employing ERs.

---

> > > ### Author Response · Authors · 2024-11-26
> > >
> > > Thank you for the comment.
> > >
> > > ## Response to Q1 (Motivation for proposing a new metric)
> > >
> > > In fact, the metric we use is a simplified version of CodeBLEU (data flow scores removed), and we do not intend to propose a new metric. As an early work, we conducted experiments on a statement-level dataset - CoNaLa. These Python statements are relatively simple, and some of them do not even have data flow. On these statements without data flow, even if the reconstructed statement $c'$ is correct, the CodeBLEU between the two statements is less than 1 because the data flow score is 0. This will mislead our framework to revise ERs that are already correct continuously. Therefore, we removed the data flow score and only considered n-gram and syntax similarity. Our preliminary experiments also showed that on the CoNaLa dataset, our metric can reliably measure the semantic similarity between statements.
> > >
> > > ## Response to Q2 (The correctness of ERs)
> > >
> > > Thank you for your suggestion. During the experiment, we manually checked the correctness of each ER and obtained eight findings based on correct ERs only. The dataset we used contained 477 code statements in total, and our framework generated correct ERs for 399 of them, with an accuracy of 83.65%. For comparison, we manually checked the ERs initially generated by LLMs (the output of the first trial), of which only 210 code statements had correct ERs, with an accuracy of 44.03%. The improvement from 44.03% to 83.65% shows the effectiveness of our framework.
> > >
> > > ## Response to Q3 (In-depth Analysis)
> > >
> > > Thank you for your suggestion. This paper is the first to discuss the ERs of code. We not only summarize the common types of ERs of code, but also analyze how large language models understand the code based on these common ERs (Findings 2, 3, and 4). These findings facilitate the interpretation of the reasoning mechanism inside large language models.
> > >
> > > Meanwhile, we recognize that the direction you suggested is reasonable (i.e., which type of ERs the models generate for which type of code). As we mentioned in Finding 5, we have found two rules: (1) When the code contains search operations (e.g., traversing an array to search for elements that satisfy the conditions), the models tend to generate SQL-style ERs; (2) When the code involves numerical operations (e.g., summation, multiplication), the models often generate arithmetic expressions. Because it is time-consuming to check ERs and summarize the rules manually, we will invest more energy in the future to provide a more in-depth analysis.
> > >
> > > ## Response to Q4 (Future Work)
> > >
> > > Thank you for your suggestion. Because ERs have wide application scenarios, we describe future directions from a high-level perspective. Next, we elaborate on the potential of our work using code translation as an example.
> > >
> > > Code translation aims to convert input code into code in a target language, such as Python-to-Java. Given a code snippet, our framework treats the code in the target language as an ER. Then, we add constraints that the text and grammar of generated ERs must satisfy the target language, for example, they must be successfully parsed by the grammar parser of the target language. Then, our framework can unsupervisedly convert the input code into code in the target language and ensure that the converted code satisfies the lexical and grammatical rules of the target language. As more and more new programming languages emerge, our framework can effectively migrate legacy software to more advanced languages.

---

> > > > ### Comment · Reviewer_3DjZ · 2024-11-29
> > > > **Response to Authors**
> > > >
> > > > I am writing to acknowledge that I have read the responses from the authors. I still decided to keep my score. Only providing statistics to my example questions for in-depth analysis is not enough.

---

### Meta-Review · Area_Chair_NCkN · 2024-12-16

**Metareview:**

This paper presents an empirical study investigating the ability of large language models (LLMs) to generate Equivalent Representations (ERs) of code. Using the Reflexion framework, the authors prompt LLMs to generate ERs in both open and constrained scenarios and analyze the resulting representations. Their findings suggest that LLMs are capable of producing ERs, providing potentials for further exploration of ER generation to support various code-related tasks.

The motivation for focusing on ER generation is not clearly articulated, and while potential implications are discussed, they remain somewhat speculative. Additionally, the evaluation metrics—semantic-equivalent score and constrained score—do not necessarily capture the functional correctness of the generated ERs. This leaves uncertainty about whether the generated representations are truly equivalent in practice.

**Additional Comments On Reviewer Discussion:**

The authors provides qualitative examples during rebuttal, but the number of examples are limited.

---

### Decision · Program_Chairs · 2025-01-22

Reject